# Implications of a ‘Third Signal’ in NK Cells

**DOI:** 10.3390/cells10081955

**Published:** 2021-07-31

**Authors:** Mohamed Khalil, Dandan Wang, Elaheh Hashemi, Scott S. Terhune, Subramaniam Malarkannan

**Affiliations:** 1Laboratory of Molecular Immunology and Immunotherapy, Blood Research Institute, Versiti, Milwaukee, WI 53226, USA; mokhalil@mcw.edu (M.K.); danwang@mcw.edu (D.W.); Ehashemi@mcw.edu (E.H.); 2Department of Microbiology and Immunology, Medical College of Wisconsin, Milwaukee, WI 53226, USA; 3Department of Pediatrics, Medical College of Wisconsin, Milwaukee, WI 53226, USA; 4Department of Medicine, Medical College of Wisconsin, Milwaukee, WI 53226, USA

**Keywords:** NK cells, Natural Killer cells, NCR—Natural killer cell receptor, IL—interleukin, IFN-γ—interferon gamma, MHC—major histocompatibility complex

## Abstract

Innate and adaptive immune systems are evolutionarily divergent. Primary signaling in T and B cells depends on somatically rearranged clonotypic receptors. In contrast, NK cells use germline-encoded non-clonotypic receptors such as NCRs, NKG2D, and Ly49H. Proliferation and effector functions of T and B cells are dictated by unique peptide epitopes presented on MHC or soluble humoral antigens. However, in NK cells, the primary signals are mediated by self or viral proteins. Secondary signaling mediated by various cytokines is involved in metabolic reprogramming, proliferation, terminal maturation, or memory formation in both innate and adaptive lymphocytes. The family of common gamma (γc) cytokine receptors, including IL-2Rα/β/γ, IL-7Rα/γ, IL-15Rα/β/γ, and IL-21Rα/γ are the prime examples of these secondary signals. A distinct set of cytokine receptors mediate a ‘third’ set of signaling. These include IL-12Rβ1/β2, IL-18Rα/β, IL-23R, IL-27R (WSX-1/gp130), IL-35R (IL-12Rβ2/gp130), and IL-39R (IL-23Rα/gp130) that can prime, activate, and mediate effector functions in lymphocytes. The existence of the ‘third’ signal is known in both innate and adaptive lymphocytes. However, the necessity, context, and functional relevance of this ‘third signal’ in NK cells are elusive. Here, we define the current paradigm of the ‘third’ signal in NK cells and enumerate its clinical implications.

## 1. Introduction

Natural Killer (NK) cells protect hosts by eliminating pathogens and transformed malignant cells. NK cells are the major subgroup of innate lymphoid cells (ILCs) [1]. The existence of this non-T cell subset was observed in the early 1960s [2,3]. NK cells were formally characterized as an independent lymphocytic lineage in the mid-1970s [4,5,6,7]. NK cells differentiate ‘*self*’ healthy cells against the infected or transformed cells through unique mechanisms. The healthy cells, by virtue of expressing host MHC class I molecules on the cell surface are defined as ‘*immunological self*’, which are recognized by inhibitory human killer cell immunoglobulin-like receptors (KIRs) or murine Ly49 receptors initiating negative signaling that trigger membrane-proximal phosphatases to block any activation effectively. Virus-infected cells often down-regulate MHC class I molecules to avoid the attack from CD8^+^ T cells [8]. Nevertheless, those cells are particularly vulnerable to NK cell-mediated clearance and are considered the immunological ‘*missing-self*’ [9]. Viral infections often lead to the expression of ligands that are typically absent on healthy cells [10]. These stress-induced ligands serve as the ‘*danger signals*’ and interact with activating receptors on NK cells to overrule the negative signaling initiated by MHC class I molecules, the immunological ‘*induced-self*’ [10]. In recent years significant progress has been made in understanding the molecular basis for these distinct mechanisms operated by NK cells. Recent discoveries have also started to reveal the critical role of NK cells in anti-inflammatory responses, tissue repair, and autoimmunity [11]. Irrespective of this progress, a critical knowledge gap exists in defining the basic biology of NK cells that is essential to realize their clinical advantages. One of these gaps relates to distinctive mechanisms of NK cell activation.

NK cells possess strong functional similarities to their distant CD8^+^ T cell cousins. NK cells mediate granzyme/perforin, FasL, and antibody-dependent cell cytotoxicity (ADCC). They also produce significant quantities of pro-inflammatory cytokines or chemokines such as IFN-γ, GM-CSF, TNF-α, CCL3, CCL4, and CCL5. However, unlike CD8^+^ T cells, NK cells can mediate cytotoxicity without prior exposure to antigens, termed ‘priming’ [4,7]. In addition, NK cells use non-clonotypic receptors as their primary signal conduits for activation. This inherent ability of NK cells to utilize non-clonotypic receptors to recognize protein ligands on target cells requires tightly balanced activating and inhibitory signaling pathways. However, we have yet to define how NK cells diverge from T cells in signaling requirements in this context. Additionally, NK cells communicate with dendritic cells (DCs), macrophages, neutrophils, and T cells through direct cell-cell interactions or soluble factors (Figure 1).

Activation via TCR, BCR, or non-clonotypic receptors such as NCRs, NKG2D, and Ly49H is ‘primary signaling’ that depends on Src kinases and PLC-γ-based pathways. Costimulatory signals via cytokine receptors (IL-2, IL-15) and immune-checkpoint proteins (CD19, CD28, NKG2D) form the basis for the ‘second signal’, which is essential to recruit and activate PI(3)K-based signaling. Other modes of NK cell activation may involve soluble factors such as PDGF-D [12] that binds to NCR2 or cell-free ligands of NKG2D [13]. Activation through the receptors for Type I IFN-α/β, IL-12, IL-18, IL-21, IL-23, IL-27, IL-35, and IL-39 form a distinct set of signaling that constitutes a ‘third signal’. This distinct activation set largely depends on STATs and TRAFs as the conduits of signal transduction [14]. STAT4 downstream of IL-12 is involved in chromatin remodeling, thus providing a distinct definition and necessity for the ‘third signal’ [15]. While the requirement for the ‘third signal’ is established in NK cells, its molecular basis is yet to be fully understood. Here, we describe the molecular mechanism and the clinical relevance of ‘primary’, ‘secondary’, and ‘third signal’ that independently operate on NK cells.

## 2. Germline-Encoded Receptors form the Conduits of ‘Primary Signal’ in NK Cells

NK cells are the major member of the innate lymphocytes and are essential in the early immune response against viral infections and tumor clearance [16,17,18,19]. NK cells require ‘primary signals’ through their non-clonotypic germline-encoded activating receptors such as NKG2D, Ly49H, and NCR1 to execute their effector functions. Compared to clonotypic receptors, the ‘primary signals’ received through these NK receptors are distinctive because they can initiate a high threshold immune response without prior stimulation upon recognizing virus-infected and stress-induced cells. NKG2D, Ly49H, NCR1, and NCR3 initiate ‘primary signaling’ when interacting with stress-induced self-ligands on tumor cells or viral proteins expressed on infected cell surfaces. Figure 2A,B present the major cell surface receptors expressed on murine and human NK cells, respectively. The ‘primary signal’ and its functional outcomes, including the release of cytokines and cytotoxic granules, are determined by the delicate balance between activating and inhibitory signaling pathways [20]. The inhibitory receptors tightly regulate the progressive threshold of ‘primary signals’, thereby avoiding a catastrophic autoimmune response. In a practical sense, the inhibitory signaling pathways augments the threshold of activating ‘primary signals’ required for gene transcriptions and mediating effector functions. Apart from the germline-encoded receptors, integrins play an essential role in executing the effector functions of NK cells [21]. Integrins, including LFA-1 and its ligand ICAM-1, are integral parts of the NK cell immunological synapse (NKIS) and their absence leads to impaired cell-cell interaction and significantly reduced target cell lysis [22]. Importantly, activation via NKG2D initiates an ‘inside-out’ signaling that activates resting LFA-1 and increases its quantity in the plasma membrane by exporting them from the intracellular stores [23,24]. Thus, the intimate cooperation between activation receptors and integrins is crucial for the successful execution of ‘primary signals’ in NK cells.

### 2.1. Stress-Induced Ligands Provide ‘Primary Signals’ via NKG2D

NKG2D is a homodimeric C-type lectin-like type II transmembrane glycoprotein expressed on most murine and human NK cells, recognizing stress-induced ligands structurally related to MHC class 1 (Figure 3) [25]. NKG2D is expressed by NK cells, CD8^+^ T cells [26], γδ T cells [27], and NKT cells [28]. ‘Primary signaling’ mediated via NKG2D requires the adapter proteins DAP10 and DAP12 in mice and DAP10 in humans. DAP10 activates the PI(3)K-p85α subunit, while DAP12 recruits ZAP70 and Syk leading to PLC-g2 activation (Figure 3) [29]. Stimulation via NKG2D and other activation receptors require a threshold that need to overcome the inhibitory signals mediated by KIR2DL1/2/3 and NKG2A. Ligation of NKG2A or NKG2B to HLA-E induces tyrosine phosphorylation of immunoreceptor tyrosine-based inhibitory motif (ITIM) to suppress human NK cell activation [30]. Thus, an ‘altered balance’ between the activating and inhibitory receptors forms the basis for NK cell-mediated cytotoxicity and cytokine production [31].

NKG2D ligands are expressed in both malignant and infected cells. Murine NKG2D ligands belong to non-classical MHC Class-I and fall into three sub-families, Rae1 [32], H60 [33], and Mult1 [34]. All three sub-families are clustered on Chromosome #10 [35]. The H60 sub-family has three members (a, b, and c), and the first identified member, Histocompatibility antigen 60a (H60a), is a minor histocompatibility antigen [36,37]. H60a and H60b possess transmembrane and cytoplasmic domains, while H60c uses a GPI anchor. Retinoic acid-induced early transcript (Rae-1) has five members (α-ε), and they were identified as responders to retinoic acid [38]. All five members of the Rae-1 sub-family are attached to the plasma membrane through GPI anchors. The murine ULBP-like transcript (Mult1) has a single member and is distantly related to the other two sub-families [39]. Mult1 possesses the most extended cytoplasmic domain. All nine members of the murine NKG2D ligands have an α2 extracellular domain but do not contain the canonical α3 domain, thus lack the ability to bind to b-2 microglobulin and present antigenic peptides. In humans, NKG2D ligands are in chromosome 6, which are grouped into two families, MHC class I chain-related protein (MIC) and UL16-binding protein (ULBP). MIC has two members, MICA [40] and MICB [41]. The ULBP family has ten genes, of which five are transcribed. Cellular stress conditions such as oncogenic proliferation, viral infection, or hypoxic microenvironment upregulate ligands for NKG2D [42].

### 2.2. Primary Signals’ from Viral Ligands Activate via NCR1 and Ly49 Receptors

Three NK cell receptors (NCR1, NCR2, and NCR3) are the main activating receptors of human NK cells [43,44,45]. The stimulatory receptor NKp46/NCR1 was the first identified NCR [46,47]. It efficiently triggers the release of cytotoxic granules, cytokines, and chemokines upon binding ligands of viral [48], bacterial [49], and cellular origin [50]. Mice express only NCR1, which is an ortholog of NKp46 (also termed MAR-1) [51]. NCRs are natural cytotoxic receptors and type 1 transmembrane glycoproteins characterized by C2-type immunoglobulin-like domains and are a major activator in human NK cells [44,52,53]. Upon activation of NCRs, there is increased Ca^2+^ mobilization, increased NK cell cytotoxicity, and augmented cytokine production [54]. 

In mice, like NKG2D, Ly49 is a group of homodimeric receptors that are C-type lectin-like type II transmembrane glycoprotein that binds to MHC Class-I molecules in a peptide-dependent but not peptide-specific manner [55]. Unlike NKG2D, this group of receptors in mice can be activating or inhibitory, and it is the summation of signals from activating and inhibitory Ly49 receptors that ultimately determine the development and function of NK cells. Tyrosine phosphorylation of an ITIM is restricted to inhibitory Ly49 receptors leading to Src homology 2 domain-containing protein tyrosine phosphatases recruitment [56].

NK cells mediate essential anti-viral immunity, especially to the herpesvirus family [57]. Cytomegalovirus (CMV) represents a well-studied viral infection model for both human and murine NK cells. NK cell depletion and adoptive transfer experiments have established the anti-viral function of NK cells against CMV in mice [58]. Murine CMV (MCMV) infection leads to the expression of the ligand m157, which is an MCMV-encoded glycoprotein recognized by the activating Ly49H receptor on murine NK cells (Figure 4) [59]. Recent works have demonstrated a unique adaptive feature of this antigen-specific germline-encoded receptor [60], allowing for cytolytic killing and clonal expansion of Ly49H^+^ NK cells (Figure 4). Unlike inhibitory receptors, activating Ly49 receptors are not phosphorylated upon activation. The Ly49H receptor is expressed on a subset of NK cells and utilizes the DAP12 adapter protein. This receptor recognizes the MHC Class-I-like viral protein, m157, expressed by the MCMV [61]. NK1.1 is similar to Ly49H, and instead of signaling via DAP12, it signals FcRe1-gamma through immunoreceptor tyrosine-based activation motif bearing [62]. Multiple other activation receptors are known to mediate ‘primary signaling’ in NK cells that are described in detail elsewhere [52].

This ‘primary signal’ activates NK cells and initiates the production of large amounts of pro-inflammatory cytokines and direct cytolytic clearance of the virus-infected cells [63,64]. Murine strains which lack Ly49H-expressing NK cells are susceptible to a more severe MCMV infection [65]. NKG2C recognizing the HLA-E loaded with an HCMV viral peptide is presumably the human counterpart of the m157-Ly49H axis [66,67]. A similar immunosurveillance mechanism has been found in influenza infections where the virus-encoded hemagglutinin is recognized by another NK cell activating receptor, NCR1 [68]. Besides direct control of viral infections, NK cells also shape the adaptive immune response [69]. The pro-inflammatory cytokines, especially IFN-γ produced by NK cells, regulate the differentiation of Th1 cells, a critical T cell subset that controls viral infections [70].

### 2.3. Fc Receptors and NK Cell Activation

ADCC represents a classical example of cell-cell collaboration between NK and B cells. ADCC is an Fc-dependent effector function of IgG that plays an essential role in anti-viral and anti-tumor immunity. Immunoglobulins (Igs) are the most efficient soluble factors of the humoral immunity. In addition, Igs also facilitate cell-mediated cytotoxicity through ADCC. The ‘Fraction absorbable’ (Fab) recognize either free pathogens or antigens on the surface of the infected or malignant cells. The ‘Fraction crystallizable’ (Fc) possess the ability to bind to several receptors (FcR). The low-affinity FcγRIIIA (CD16A) expressed by human CD56^dim^ NK cells has an extracellular domain recognizing the Fc region of antigen-specific antibodies, which binds to ligands expressed on malignant cells [71].

ADCC by NK cells is mainly mediated by IgG-subclasses IgG1 and IgG3 through CD16A. Upon activation of CD16A, immunoreceptor tyrosine-based activation motifs (ITAM) are phosphorylated by tyrosine kinases (ZAP-70 and Syk), which initiates murine NK cell effector cytokine production and granule exocytosis [72]. In both humans and mice, NK cell-mediated ADCC occurs through three pathways, (1) proinflammatory cytokine release (IFN-γ); (2) exocytosis of cytotoxic granules, and (3) TNF family death receptor signaling. One mouse study revealed co-administration of tumor-specific monoclonal antibodies and IL-15 led to enhanced NK cell-mediated ADCC [73]. In humans, multiple studies have demonstrated the significant clinical potential of NK cell-mediated ADCC in clearing tumors [74]. These studies emphasize ways clinicians can better utilize NK cells for effective cancer immunotherapies.

### 2.4. Innate Receptors and NK Cell Activation

The last set of receptors capable of mediating ‘primary signals’ are the pattern recognition receptors (PRRs) [75,76]. The mechanism of TLR-mediated activation in NK cells remains elusive [45,77]. Adaptor protein MyD88 associates with all TLRs except TLR3. TLR2 expressed on human NK cells recognize cell wall components of *Mycobacterium tuberculosis* [78] and *Mycobacterium bovis* [79]. The absence of TLR3 in NK cells resulted in augmented experimental B16F10 lung metastasis in mice [80]. Activation of human NK cells via TLR4 with lipopolysaccharide resulted in IFN-γ production [81]. TLR5 expressed on human NK cells recognizes bacterial flagellin [82]. TLR7 and TLR8 expressed on human NK cells recognize uridine-rich ssRNA derived from the HIV-1 long terminal repeat [83]. Stimulation of human NK cells via TLR9 by baculovirus resulted in IFN-γ production and upregulation of CD69 [84]. NK cell-mediated direct killing of bacteria-infected cells relies on delivering granzymes and initiation of cell death program in the target cells [85]. IFN-γ production from NK cells is critical for the early control of *Listeria monocytogenes* infection [86,87]. Moreover, NK cells are involved in several other microbial infections, including *Staphylococcus aureus*, *Lactobacillus johnsonii*, *Mycobacterium tuberculosis*, and *Mycobacterium bovis* bacille Calmette-Guérin (BCG) [85,88]. In addition to bacterial infections, NK cells also contribute to the anti-fungi immune response through various mechanisms, including perforin-mediated destruction of fungal membranes, direct phagocytosis, and production of pro-inflammatory cytokines [89,90]. The essential role played by NK cells was demonstrated in both human NK cell deficiency and genetically-modified mouse models [57,58]. Individuals with loss of NK cells’ functions suffer from reoccurring herpes viral infections [57]. The two effector functions of NK cells are the production of inflammatory cytokines and cytotoxicity, both of which profoundly impact immune responses against pathogens [52,91].

Costimulatory molecules form another set of receptors that can mediate secondary signals, which display remarkable diversity in expression, function, and structures. NKR-P1 is a type II glycoprotein receptor of the C-type lectin family and acts as both a costimulatory and co-inhibitory receptor [92]. Like the Ly49 family, the inhibitory signaling is mediated via ITIM while the activating signaling is via ITAM [93]. TIGIT family receptors include a cluster of immunoglobulin superfamily receptors that interact with Nectin and nectin-like molecules and have recently received notable recognition as potential cancer immunotherapy targets [94]. These receptors include TIGIT (T-cell immunoglobulin and ITIM domain), DNAX accessory molecule-1 (DNAM-1, also known as CD226), T cell activated increase late expression (TACTILE, also known as CD96), and PVR-related Ig domain (PVRIG also known as CD112R). DNAM1 is a cell surface glycoprotein functioning as an adhesion molecule synergizing with activating receptors to trigger NK cell-mediated cytotoxicity [95]. DNAM-1 recognizes the poliovirus receptor or CD155 and CD112 (nectin-2) on tumor cells which induce NK cell-mediated lysis [96]. Additionally, costimulatory molecules CD80, CD86, and CD40 assist in NK cell-mediated tumor cell lysis [97,98].

### 2.5. Molecules Associated with ‘Primary Signals’ in NK Cells

Recent studies revealed two distinct primary signaling pathways downstream of activation receptors [99,100,101,102,103]. These two pathways formed the basis for signaling requirements for cytotoxicity and inflammatory cytokine production in NK cells. Lck/Fyn→PI3K-p85α/p110δ→Itk→PLC-γ2 results in the activation of ERK1/2 and JNK1/2, which is primarily responsible for NK cell-mediated anti-tumor cytotoxicity. In contrast, the Lck/Fyn→ADAP→Carma1→Bcl10→TAK1 pathway that regulates the activation of NF-κB and AP1 is responsible for inflammatory cytokines production. Fyn and Lck are the two non-receptor pleiotropic tyrosine kinases (SFKs) that are essential to initiate signaling downstream of activating receptors. SFK family contains nine members (Src, Fyn, Lyn, Lck, Fgr, Yes, Hck, Blk, and Frk) [104]. SFKs participate in a myriad of cellular functions [105] including growth, differentiation, migration, adhesion, and in lymphocytes, cytotoxicity, and cytokine production [106]. In NK cells, following activation, Fyn is trans-phosphorylated by Lck and recruits signaling partners PI(3)K-p85α [107] and the lymphocyte-specific scaffold protein Adhesion and Degranulation-promoting Adaptor Protein (ADAP) [108]. Recruitment of these two substrates through Src-homology (SH) domains allows Fyn to initiate and propagate distinct effector functions. A lack of Lck did not alter the NK cells development, stimulation with poly (I:C), or IL-2 [109]. However, knockdown of Lck in NK cells resulted in significant reductions in NKG2D- and CD137-mediated cytotoxicity and cytokine production [101]. Primary signaling in NK cells utilizes PLC-γ2 to generate second messengers [110]. PLC-γ is the major regulator of intracellular calcium and works by hydrolyzing phosphatidylinositols (PIP2) into diacylglycerol (DAG) and inositol 1,4,5-triphosphate (IP3) [111].

A signaling cascade involving ADAP is obligatory for the production of proinflammatory cytokines. Fyn using its SH2 domain recruits ADAP [112]. Lack of ADAP did not affect the NK cell-mediated anti-tumor cytotoxicity; however, was essential for the production of IFN-γ, GM-CSF, TNF-α, MIP-1α, MIP-1β, and RANTES. Membrane-bound Fyn/ADAP complex recruits downstream signaling proteins Carma1 and TAK1, which leads to phosphorylation and degradation of IκBα and nuclear translocation of NF-κB [113]. Independently, ADAP also recruits TAK1, which facilitates the phosphorylation of IKKα and IKKβ [113], components of the NF-κB signaling pathway. Thus, ADAP contributes to CBM complex formation in response to ITAM-containing receptors [100,114,115].

## 3. Common-Gamma Receptors Transmit ‘Second Signals’ in NK Cells

Most NK cells develop and terminally mature in the BM [116]. Common lymphoid progenitors (CLPs) develop into early innate lymphoid progenitors (EILPs). T, B, three ILC lineages, and the conventional NK cells originate from EILPs [117]. NK cell development depends on multiple common-gamma chain-containing cytokine (γc, CD132) receptors (IL-2, IL-4, IL-7, IL-15, and IL-21) that utilize PI(3)K-based signaling pathway [118]. The lineage-committed NKPs express IL-15/IL-2 receptor β chain (CD122) [119]. The signaling divergence among the γc chain receptor family has not been fully defined; however, it depends on the unique α-chain utilization and the differential activation of unique STATs.

Common γ cytokines are required to regulate the initiation, amplification, and maintenance of transcriptional memory following immune responses [120]. These cytokines also play a crucial role in establishing ‘immunological priming’. This phenomenon has been well-documented for B and T cells and forms the basis for adaptive immunity [121]. The critical effects of γc cytokines in modulating NK cell responses are studied intensively. The common gamma receptor (γcR)-interacting cytokines IL-2, IL-7, IL-15, and IL-21 are being extensively used to stimulate and expand NK cells for adoptive transfer in the clinical setting [122]. Unique α-chains define the receptors for these cytokines (Figure 5). IL-2 and IL-15 share a β-chain and the γcR along with cytokine-specific IL-2Rα and IL-15Rα, respectively [123,124], and transduce the signals primarily via PI(3)K [125]. Historically, IL-2 has been extensively used in vitro to expand murine and human NK cells [126]. IL-15 activates the PI(3)K-mediated mTORC1 pathway (Figure 5) [127]. In contrast to the defined functions of IL-4 in skewing T cells to a Th2 functional phenotype, it augments [128]. Also, the addition of IL-12 and IL-2 along with IL-4 increased the production of IFN-g in NK cells [129]. The role of IL-9 in NK cell development and function is currently not known. Although IL-2, IL-15, IL-12, and IL-18 are widely used to expand and activate NK cells, their in vivo role in coordinating the activation via receptors such as NKG2D or NCR1 is not understood.

### 3.1. IL-2: Context Unknown

IL-2 is primarily produced by CD4^+^ T cells. In the spleen, the majority of the NK cells are located within the red pulp area and not inside the germinal centers where CD4^+^ T and B cells organize [23]. NK cells are predominantly present in the subcapsular region during *Toxoplasma gondii* infections in lymph nodes, where they interact with DCs or macrophages [130]. Thus, a context in which NK cells are proximal to CD4^+^ T cells in these secondary lymphoidal organs appears unlikely. However, this may still occur at the site of infection or inflammation. In humans, γc chain mutations result in severe immunodeficiency with nearly complete loss of NK cells [131]. IL-2-deficient mice have expected NK cell numbers and development [132] with minimal effect on their functional capabilities [133]. This demonstrates that IL-2 does not play an obligatory role in the development and functions of NK cells. IL-2 binds to the IL-2 receptor alpha chain, CD25 [134], or the IL-2 receptor beta chain, CD122, with relatively low affinity (K_d_ between 10^−8^ and 10^−6^ M), and binds with a higher affinity to a CD25/CD122 complex. However, the highest affinity receptor for IL-2 is a trimeric complex of IL-2Rαβγ [135]. IL-2R signaling is dependent on receptor composition. JAK1 associates with IL-2Rβ and JAK2 with γc [136,137]. The binding of IL-2 to the IL-2R complex activates the PI(3)K-mTOR axis and the MAPK pathways, and phosphorylation of STAT1, STAT3, and STAT5 (mostly STAT5) [138], resulting in activation signaling [139]. In NK cells, IL-2 is a potent driver of proliferation [140] and primes them for IL-12-mediated stimulation [141]. Combined stimulation with IL-2 and IL-12 increased tumor cell lysis and NK cell perforin production [142]. In human NK cells, IL-2 and other pro-inflammatory cytokines alter the repertoire of NK cell surface receptors [143].

### 3.2. IL-15: A Metabolic Re-Programmer

IL-15 is a crucial cytokine required for NK cell development, survival, and proliferation. IL-15R consists of three subunits, and the absence of any of its subunits results in failed NK cell development in mice [144,145]. IL-15 binds to IL-15Rα (CD215) alone with a high affinity [146]. In contrast to IL-2, IL-15 is obligatory for NK cell lineage commitment and maturation [147]. Also, a low dose of IL-15 is sufficient to sustain survival signaling in NK cells, while a high dose of IL-15 promotes NK cell proliferation and augments the expression of effector molecules such as perforin and granzymes [148]. Knocking out genes encoding cytokines, or the common g receptors strongly supports the notion that IL-15 is essential for the development of NK cells. For example, a lack of CD132 in mice results in the complete absence of mature NK cells [149]. Deletion of *Il15* or *Il15ra* genes but not *Il2* results in a similar loss of mature NK cells seen in the γc chain-deficient mice, establishing the central role of IL-15 in the development of NK cells [145]. Both IL-15 and IL-15Rα can be expressed in the same cell leading to the loading of IL-15 onto IL-15Rα in the ER of the cell. This high-affinity interaction results in membrane-bound IL-15/IL-15Rα complexes [146]. Thus, the IL-15 is trans-presented by cells such as DCs to the neighboring NK cells expressing the IL-2/15Rβ/γc complexes [150]. It is also interesting to note that the IL-15/IL-15Rα complex has the ability to trans-interact with DAP10 and utilize novel synchronized trans-signaling pathways linking Jak3/Stat5 to DAP10/PI(3)K-p85α/Grb2 [151]. This observation provides credence to the argument that the ‘primary’ and ‘second’ signals can be temporally integrated during the activation of NK cells.

The IL-15R activates multiple intracellular signaling pathways, including JAK1/3 dependent phosphorylation of STAT5 and PI-3K-Akt-mTOR-dependent phosphorylation of the ribosomal protein S6 by the p70-S6 kinase (Figure 5) [152,153]. The role of mTOR in the IL-15-dependent developmental progression and priming of NK cells are well-established [154,155]. IL-15R-mediated signaling includes Jak1/3-Stat5a/b, the PI(3)K-mTOR, and the MAPK pathways [156]. Cytoplasmic tails of the IL-2/15Rb/γc complex recruits Jak1 and Jak3 [157]. Jak1 and Jak3 phosphorylate Tyr392 and Tyr510 at the H-region serve as critical docking sites for Stat5a and Stat5b [158]. The absence of Jak3 significantly impairs the development and maturation of murine NK cells [159]. Also. Stat5a and Stat5b are essential for maintaining the homeostasis of NK cell numbers and complexity. In mice, Stat5b deficiency results in a significant loss of NK cells [160].

mTOR-mediated metabolic reprogramming and initiation of unique protein translation is the second major pathway downstream of the IL-15R that operates via the PI(3)K-axis. The regulatory subunit PI(3)K-p85α and its spliced smaller isoforms p55α and p50α recruit catalytic subunits p110α, p110β, and p110δ [161]. Both p110β and p110δ are obligatory for the development and functions of NK cells [162,163]. The PI(3)K-mTOR axis is essential for the development and functions of NK cells [148]. mTORC2 acts as a link between PI(3)K and mTORC1. The PtdIns(3,4,5)P_3_ generated by PI(3)K interacts with the pleckstrin homology (PH) domain of mSin1 and releases its inhibition of the kinase domain of mTORC2 [164]. Thus, PIP_3_ plays an essential role in activating mTORC2. Importantly, Akt, recruited by the membrane-bound PIP_3,_ phosphorylates mSin1 at the Thr86 site and promotes the activation of mTORC2 [165]. Akt is also a well-characterized target of mTORC2 [166], and the phosphorylated form of Akt is the major link to the activation of mTORC1 [166].

### 3.3. IL-21: Is It More than Additive?

Modulation by the pro-inflammatory type I cytokine, IL-21, mediates NK cell activation via similar receptor complexes. TfH and Th17 cells produce IL-21 [167]. IL-21 binds a heterodimer of IL-21R and γc [168]. Similar to signaling via IL-12R, JAK1 is associated with the IL-21R, and JAK3 associates with γc [169]. This signaling cascade results in the proliferation of NK cells and the transcription of IFN-γ. Costimulation with IL-2 and IL-15 increases NK cell cytotoxicity [170]. In human NK cells, adding IL-21 to cells co-cultured with antibody-coated breast cancer cells resulted in pro-inflammatory cytokine production through reverse ADCC [171]. Alternatively, treatment of NK cells with IL-21 combined with IL-15 reduced NK cell expansion and viability [172]. Regulation of NK cell functions by IL-21 remains an active research area for potential use in combination cancer therapies.

## 4. Why Are ‘Third Signals’ Unique for NK Cells?

Pro-inflammatory cytokines produced by myeloid-derived cells promote NK and T cell activation. Myeloid cell-derived cytokines such as IL-12, IL-15, IL-18, IL-23, IL-27, IL-35, and IL-39 help coordinate the receptor-mediated activation of NK cells [173]. The intricate temporal relationship between these cytokines and activation receptor-mediated stimulation of NK cells, in vivo, remains a paradox. An operational ‘third signal’ mechanism has been shown for T cells [174]. However, recent work has shown that IL-12 and IL-23 are differentially activating T and NK cells, implicating that the ‘third signal’ for these two major cell types is mechanistically distinct and divergent [175]. Yet another complexity is the unique ‘third signals’ that are required by the tissue-resident NK cells. NK cells in murine thymi depend on IL-7 and not IL-15 or IL-2 [176]. In mice, these Gata-3/IL-7-dependent NK cells express a lower level of inhibitory Ly49 receptors and potentially play an essential role in the T cell ontogeny, including the elimination of negatively selected T cells. The IL-7Ra^+^ (CD127) thymus-derived NK cells in mice (and CD56^hi^CD16^−^ in humans) in the LN produce a higher amount of IFN-g and were unable to mediate optimal levels of cytotoxicity compared to peripheral NK cells [176]. Human liver-resident CD56^bright^/CCR5^+^/CXCR6^+^/CD69^+^ NK cells constitutes 50 % of total hepatic lymphocytes [177]. These unique NK cells express CXCR3, CXCR6, and CCR5 and depend on the respective chemokines for their liver residency and functions [178]. Human lung-resident CD56^dim^CD16^+^ NK cells are predominantly terminally differentiated. In mice, a subset of innate-like NK cells express IL-23R and produce IL-22 that is needed for the regeneration of tracheal epithelial cells [179]. Similarly, NK cells in the uterine tissue are involved in placental vascular remodeling via the production of the IL-23/IL-22 axis [180,181]. These studies demonstrate the cytokine-mediated second part of ‘third signals’ in NK cells are divergent and tissue-specific.

### 4.1. DC-NK Interactions Set a Foundation for ‘Third Signal’

NK cells can be primed by a number of inflammatory stimuli. The context and the molecular basis of priming of NK cells are different from that of adaptive lymphocytes. Myeloid cells provide the predominant priming signals to NK cells. Among these, dendritic cells (DC) play a central role [182]. A complex interplay between DCs and NK cells is an obligatory step for the NK cells’ sensitization [183]. Given DC generates critical cytokines such as IL-12, IL-15, IL-18, IL-23, IL-27, and IL-35, the crosstalk with NK cells determines the pathophysiological outcome of an ongoing immune response [184]. Priming with Type-1 IFN-α/β results in the expression of IL-15Rα and generation of IL-15 from plasmacytoid DCs- a specialized subset of DCs that use toll-like receptors to recognize intracellular viral DNA/RNA [183]. Multiple cell types, including NK cells, produce Type-1 IFNs by which they can prime DCs [185]. The trans-presentation of IL-15 by IL-15Rα to IL-15Rα/IL-2Rβ/IL-2Rγ complex on NK cells initiates multiple cellular and functional outcomes, including proliferation and transcriptional reprogramming [186].

### 4.2. IL-12 Cytokine Family, Prime Mediators of ‘Third Signal’

Members of the IL-12 family consist of heterodimeric cytokines such as IL-12, IL-23, IL-27, IL-35, and IL-39 (Figure 6) [187]. IL-12 is made up of p35 and p40 subunits, and it binds to the IL-12 receptor (IL-12Rβ1 and IL-12Rβ2) [188]. IL-12R signaling is propagated by Tyrosine kinase 2 (Tyk-2)/JAK-2 and activates STAT4 [189]. Activation by IL-12 is significantly influenced by IL-18R, which does not belong to the IL-12R family. Both IL-12 and IL-18 can be concurrently produced by the DCs. Therefore, they together exert an influence on the activation of NK and other adaptive lymphocytes. IL-18 belongs to an IL-1 family that interacts with a heterodimeric receptor composed of IL-18Rα and IL-18Rβ (Figure 6) [190]. IL-12 and IL-18 enhance NK cell effector functions, including IFN-γ production [191]. However, the IL-12 and IL-18 response is acute and independent of NK cell activating and inhibitory receptors [192]. IL-23 is another heterodimeric cytokine composed of p19 and p40 subunits, and its receptor is the heterodimeric complex of IL-23Rα and IL-12Rβ1 [193]. IL-23 activates NK cells to produce IL-22 [194]. IL-12 and IL-23 are produced by pathogen-activated macrophages and DCs and share a common component of their heterodimeric receptors, IL-12Rβ1 [187]. Although the function of IL-23 in NK cells remains under debate, the role of IL-12 in NK cell activation is well-established [187]. IL-35 contains p35 and EBI3 (Epstein-Barr virus-induced gene 3) subunits, and its recently defined receptor consists of IL-12Rβ2 and gp130 [195]. gp130 is the shared receptor subunit of an IL-6 family of cytokine receptors [196]. IL-27 is another heterodimeric cytokine that belongs to the IL-12 family and consists of p28 and EBI3 [197]. The receptor for IL-27 is composed of gp130 and WSX1 [198]. IL-27 has both activating and inhibitory functions [199,200], and IL-35 is an immunosuppressive cytokine produced exclusively by regulatory T cells (Tregs) [201].

Is the ‘third signal’ susceptible to inhibition? While an activation receptor-independent and cytokine-dependent ‘third signal’ elicits effector functions in NK cells, the regulatory network that is needed to curtail this activation is unknown. In particular, the role of inhibitory KIR in human and Ly49 receptors in mice in containing the activation mediated by the IL-12R family is novel but unexplored. Stromal cells or DCs that generate the IL-12 family cytokines may also provide a regulatory signal via their MHC Class I. In this context, it is important to note that inhibitory KIRs in humans and Ly49 in mice recruit and activate PTP or SHP-1 that dephosphorylate multiple membrane-proximal substrates and transcription factors such as STAT4. Therefore, the framework has been established for the possibility that the ‘third signal’ mediated by the IL-12R family could be subjected to Thus, the IL-12R family mediated signal in NK cells may vary depending on the presence or absence of cell-cell contact-based interaction.

Apart from IL-18, signaling by IL-12 also synergizes with IL-2, and IL-15, to significantly enhance IFN-γ production by NK cells [202]. IL-18R uses signaling adaptors, myeloid differentiation primary response 88 (MyD88), and IL-1R-associated kinase (IRAK) to propagate and complement IL-12R-mediated activation [203,204]. IL-18 alone is not sufficient to induce IFN-γ production. Moreover, the expression of IL-18R is induced by IL-12-mediated activation in lymphocytes [205]. Specifically, STAT4 activation by IL-12 poises the *Ifng* locus and enhances *Ifng* gene transcription while IL-18R signaling simultaneously induces the promoter binding activity of AP-1 and activates p38 MAPK promoting *Ifng* transcript stability and IFN-γ protein translation [206,207].

IL-12 transcriptionally primes both T and NK cells [208,209]. In T cells, cytokines function as a ‘third signal’ along with the activation receptor ‘primary signal’ and costimulation ‘second signal’ [210,211]. The functionality of a ‘third signal’ on T cells has been demonstrated. IL-12 and IFNα/β are the major sources of the ‘third signal’ for CD8^+^ T cells [212], which elicit a common regulatory program involving the expression of more than 350 genes [213]. IL-12 and IL-23 are pro-inflammatory, while IL-27 and IL-35 are inhibitory. APCs such as DCs, monocytes, and macrophages produce IL-23 following activation. NK cells express the IL-23R, composed of two subunits, IL-23R and the IL-12Rβ1 [193,214]. The binding of IL-23 to the receptors initiates signals through JAK2 and STAT3 or STAT4 [215]. Both IL-12 and IL-23 are required for NK cell responses when re-challenging mice with *Toxoplasma gondii*, implying a role for both cytokines in the clearance of infection [216]. Other ways IL-23 may impact NK cell functions and whether the IL-12 family of cytokines plays an essential role in transcriptional-priming of NK cells are areas of active investigation [217].

IL-12R signaling in NK cells is crucial for NK cell responses to virus and bacterial infections. IL-12 is central for NK cell effector functions and was initially named NK cell-stimulating factor (NKSF) [218]. IL-12 has two subunits, IL-12p35 and IL-12p40, which form disulfide bonds to become IL-12p70 [219]. The IL-12 receptor (IL-12R) is a heterodimer of IL-12Rβ1 and IL-12Rβ2 and is highly expressed in NK cells [220,221]. Production of IL-12 is primarily by APCs such as DCs, monocytes, and macrophages [222,223]. It is one of the early responses to viral and bacterial detection. Binding of IL-12p70 to the IL-12R activates IL-12Rβ1-associated tyrosine kinase 2 (TYK2) and IL-12Rβ-associated Janus Kinase 2 (JAK2) [224,225], resulting in phosphorylation of STAT4 [226], and to a lesser extent, STAT1 and STAT3. STAT4 is required for IL-12-mediated production of IFN-γ and cytotoxicity in human and mouse NK cells [141,227]. The function of STAT1 and STAT3 downstream of the IL-12R remains unclear. The phosphorylation of STAT4 allows it to homodimerize and translocate to the nucleus [228,229]. Within the nucleus, STAT4 is a transcription factor stimulating the expression of IFN-γ [189], IL-10 [230], and perforin [231], and induction of T-bet [227]. In the absence of common γ cytokines, IL-12 can drive the expansion of a large number of NK cells (lymphopoiesis) [232]. There is some evidence that IL-12R signaling may play a role in the development of memory NK cells. IL-12-mediated STAT4 activation is necessary for the generation of MCMV-specific memory NK cells [233]. Treating NK cells with IL-12, IL-18, and low-dose IL-15 induced memory-like, long-lived NK cells [234]. With human NK cells, IL-12 promotes the maturation of NK cells. Low-dose IL-12 decreases the expression of CD56 and can lead to an increased number of CD56^dim^CD16^+^ NK cells [235].

The ability of IL-12, IL-18, and IL-27 to function as the ‘third signal’ to prime NK cells has been explored [213,236]. IL-12 increases *Ifng* gene expression through chromatin remodeling, relieving gene repression and allowing continued transcription by promoting augmented histone acetylation [213]. Thus, IL-12 and IL-18 promote pro-inflammatory cytokines and chemokines generation in NK and T cells [237]. Transcriptional priming of effector lymphocytes by cytokines leads to genomic imprinting, resulting in unique cell fate decisions and distinct functional outcomes. IL-12 mediates such prototypical chromatin modifications on signature cytokine genes such as *Ifng* and *Il4* in T cells primarily via signal transducer and activator of transcription 4 (STAT4)-dependent chromatin remodeling [238,239].

IL-12, in combination with IL-18, is a potent activator of NK cells. IL-18 is a member of the IL-1 cytokine family and has a regulated production. IL-18 is produced as pro-IL-18 and activated upon cleavage of the N-terminal tail by caspase-1 [240]. IL-18 is produced by DCs, macrophages, and neutrophils [241], and microglia and epithelial cells [242]. The binding of IL-18 to the IL-18R expressed on NK cells augments IFN-γ the production via signaling via MyD88. MyD88 recruits IRAK4, resulting in phosphorylation of TRAF6 and activation of NF-κB and the MAPK pathway [203]. These pathways both promote the transcription and stabilization of the *Ifng* mRNA [206]. Specifically, in NK cells, IL-18 with IL-12 is critical for IFN-γ production and protection against fungal [243], viral [244], and bacterial infections [245]. Treatment of NK cells with IL-18 increases surface expression of the high-affinity IL-2R and CD25 (IL-2Rα), making NK cells more sensitive to IL-2 signaling during infection [246]. In vivo, IL-18 stimulation of human NK cells promoted IFN-γ generation and antibody-dependent cell-mediated cytotoxicity (ADCC) to the lymphoblast-like tumor cell line Raji following treatment with rituximab, making IL-18 a potential cytokine for combination cancer therapy [247]. Overall, IL-18 is an essential cytokine in potentiating IL-12-mediated NK cell effector functions.

### 4.3. IL-23 and IL-27 Diversify ‘Third Signal’

IL-23 belongs to the IL-12 family and is composed of p40 and p19 subunits and signals through a heterodimeric IL-12Rβ1 and IL-23 receptor complex [214,248]. Like IL-12, IL-23 is secreted by monocytes, macrophages, and DC in response to viral, bacterial, and fungal antigens [249]. IL-23 binds to IL-23R and IL-12Rβ1 but not IL-12Rβ-2 and creates an autocrine loop that mediates various inflammatory mediators’ production (Figure 6) [250].

The signaling pathway of IL-23 comprises various signaling proteins and receptor chains, including Janus kinase 2 (JAK2), tyrosine kinase 2 (Tyk2), STAT1, STAT3, STAT4, and STAT5. JAK2 and Tyk2 phosphorylate and activate STAT3 and STAT4 [251]. STATs dimerize and translocate to the nucleus, resulting in the activation of target genes. IL-23 activation of NK cells contributes to their development and anti-tumor responses [252]. Additionally, IL-23 also inhibits NK cell responses and thus can act as an inhibitor [253].

IL-27 is an inhibitory member of the IL-12 cytokine family. However, there is mounting evidence that when NK cells are stimulated with IL-27, they upregulate activation markers like CD25 and CD69 and become more sensitive for IL-18 activation. Additionally, IL-27-stimulated NK cells exhibit increased cytotoxic potential against tumors in vivo and upregulated perforin and granzyme B [254]. Murine NK cells treated with IL-27 induced phosphorylation of STAT1 and STAT3, enhancing NK cell cytotoxicity in head and neck squamous cell carcinoma mouse models [255]. Other studies have demonstrated the ability of IL-27 to modulate the NK cells’ anti-tumor cytotoxicity responses [255]. Therefore, it may be beneficial to investigate further the role of IL-27 in enhancing NK cell effector functions because these studies demonstrate that IL-27 augments NK cells cytotoxic responses to a variety of tumor cell lines, including perforin, granzyme, TRAIL, and Fc-γR-III-dependent mechanisms [255]. The role of IL-27 in NK cell-mediated anti-tumor immunity has been defined [217]. However, the underlying molecular mechanism is not well-defined. Notably, the mechanism by which IL-27 regulates NK cell effector functions during viral infections is yet to be fully understood.

An additive effect was not observed when NK cells were stimulated with IL-27 and IL-12 or IL-18, suggesting a unique transcriptional and functional role of IL-27. Similarly, pro-inflammatory cytokine IL-6, which shares its receptor subunit with the IL-27 receptor, did not augment IFN-γ production either alone or along with anti-NKG2D mAb-mediated activation of NK cells. Thus, it is possible that during acute viral infection, neutrophils and monocytes/macrophages may provide the ‘third signal’ to NK cells at the site of infection through the production of IL-27. This function may be distinct from the role of dendritic cells that produce both IL-12 and IL-18 within the secondary lymphoid organs (draining lymph nodes), which provide a full-fledged activation of NK cells to mediate effector functions. Interestingly, earlier reports have suggested that IL-27 can selectively regulate NK cell subsets, suggesting the cytokines’ vital role in managing human NK cells [256]. Regardless, detailed transcriptomic and genomic analyses are required to define the temporal and independent roles of these cytokines and their producers.

The last and most recently discovered members of the IL-12 cytokine family are IL-35 and IL-39. IL-35 plays an important role in immune regulation via inhibition. The IL-35R is composed of the IL-12Rα chain p35 and IL-27β chain Ebi3. IL-35 is produced by Tregs [257] and Bregs [258] and has been shown to dampen the effector functions of Th17 cells [259]. Unlike other members of the IL-12 cytokine family (IL-12, IL-23, IL-27, and IL-39) inducing IFN-γ production by T-helper cells, IL-35 seems to function solely as an anti-inflammatory cytokine inhibiting effector T-cell proliferation and activation [195]. This same study has demonstrated that a loss of IL-35 expression leads to the less suppressive capacity of Tregs. While IL-35 plays an important role in dampening immune responses, the role of IL-35 on NK cell signaling has not been well established. The IL-39 heterodimer comprises the IL-23R chain p19 and IL-27β chain Ebi3 and is secreted by lipopolysaccharide-stimulated B-cells [260]. Like other members of the IL-12 cytokine family (IL-12, IL-23, IL-27, and IL-35), IL-39 activates STAT signaling. The role of IL-39 on NK cell signaling and activation remains unclear.

## 5. Summary and Future Outlook

Our understanding of cytokines’ roles in shaping both innate and adaptive immune systems has profoundly evolved in the past three decades. Cytokines not only act as mediators for our immune system (both positive and negative feedback), but they also work in fine-tuning the immune responses in clearing infections, tumors, and regeneration of damaged tissues. The role of cytokines in the early lineage commitment, development, and maturation of NK cells is well-established. The common gamma chain-bearing cytokine receptors primarily mediate these. The primary stimulation via activating receptors constitutes the first line of signaling reception by the mature NK cells. However, this alone does not lead to an optimal immune response from NK cells. Mature NK cells require the adjunct role of cytokines to promote expansion, reprogram metabolic pathways, and set genes poised for transcription. Apart from these, recent evidence suggests the presence of a ‘third signal’ that can initiate, promote, and sustain the effector functions of NK cells. This ‘third signal’ can mediate cytokine production and cytotoxicity independent of major activating receptors (Figure 7). The functionality of the ‘third signal’ is further augmented when combined with ‘secondary signals’ mediated by common gamma chain-bearing cytokine receptors.

While these findings provide an exciting basis for novel immunotherapeutic approaches, the reasons for their existence and the molecular basis are elusive. Are IL-12 family cytokines the only ones capable of mediating the ‘third signal’? How is the ‘third signal’ regulated by the developmental rules such as ‘licensing’ or ‘arming’? Do MHC-Class I-KIR interactions inhibit the IL-12 family-mediated activation of NK cells? Do other checkpoint proteins such as PD1, CTLA4, TIM3, LAG3, or VISTA regulate IL-12 cytokine family-mediated NK cell activation? Under what circumstances do NK cells require the ‘third signal’? Future investigations in answering these questions will provide important insight into ways we can harness the full potentials of NK cells in the clinic.

## Figures and Tables

**Figure 1 cells-10-01955-f001:**
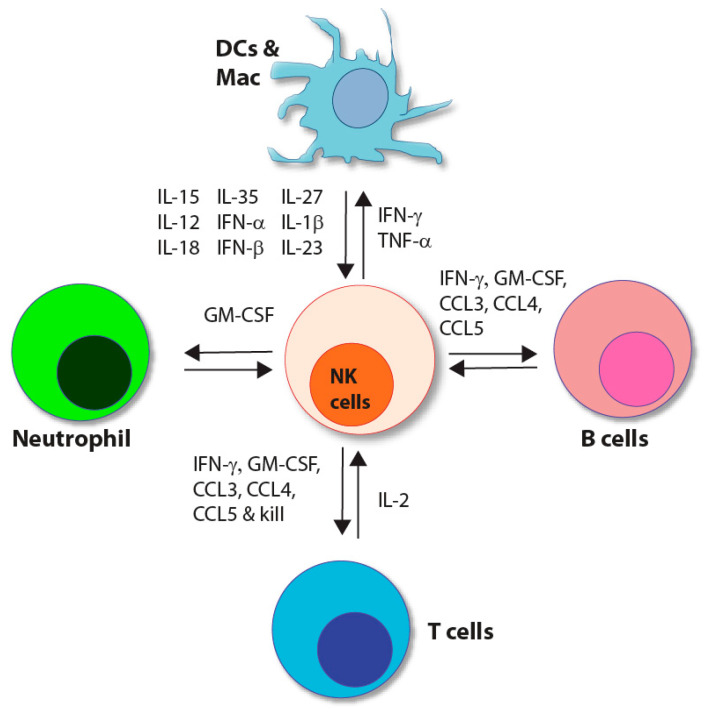
Bi-directional interaction of NK cells with other immune cells. A brief representation of NK cells’ significant interactions and the innate (myeloid-derived) and adaptive (T and B cell) immune system. Myeloid-derived cells (dendritic cells, macrophages, and neutrophils) secrete various cytokines and chemokines, regulating NK cell effector functions. NK cells secrete cytokines and chemokines (IFN-γ, TNF-a, GM-CSF, CCL-3, CCL-4, and CCL-5) to regulate lymphoid-derived cells (T and B cells).

**Figure 2 cells-10-01955-f002:**
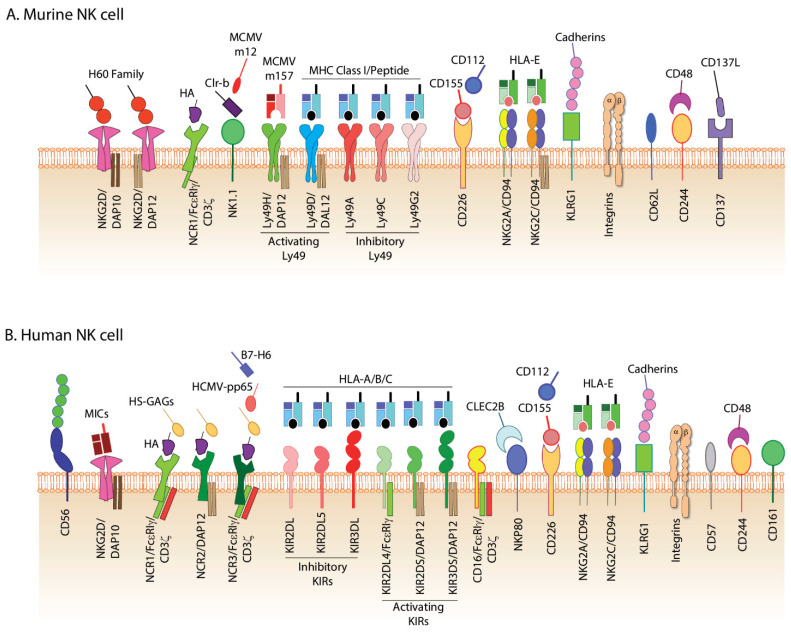
Major cell surface receptors expressed in (**A**) murine and (**B**) human NK cells. NK cells express activating receptors, inhibitory receptors, integrins, and others. HA, influenza virus-derived hemagglutinin; MCMV, murine cytomegalovirus; MHC, major histocompatibility complex; MICs. Major histocompatibility-related chains (MIC-A/B); HS-GAGs, Heparan sulfate glucosaminoglycans.

**Figure 3 cells-10-01955-f003:**
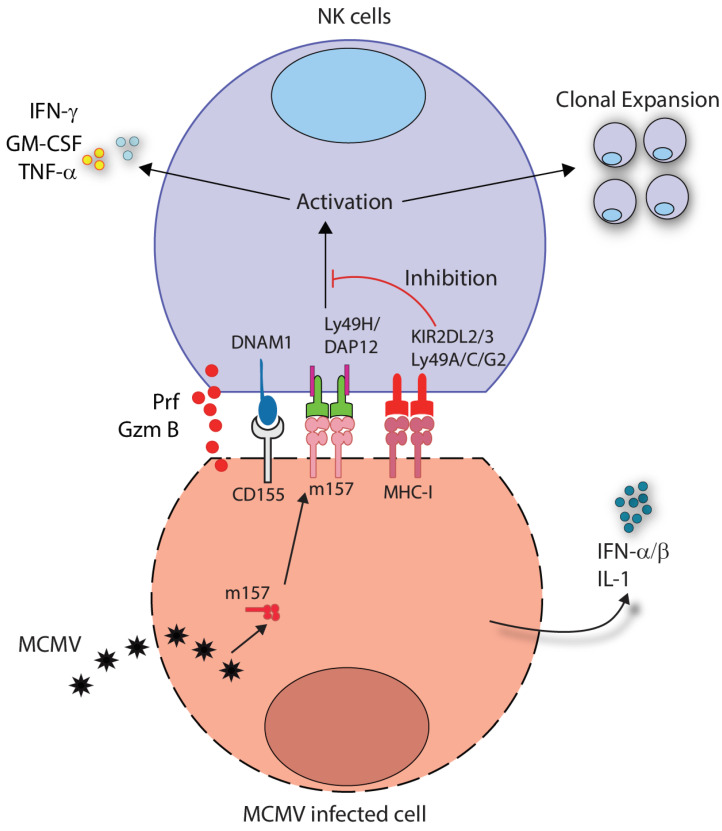
Germline-encoded Ly49H mediate viral antigen-specific recognition. During mouse cytomegalovirus (MCMV) infection, NK cells expressing the Ly49H receptor recognize MCMV-infected cells via the MCMV-derived glycoprotein m157. Stimulation of the Ly49H leads to the activation of the DAP12-mediated signaling pathway, driving NK cell degranulation and secretion of perforins and granzymes. DNAX accessory molecule-1 (DNAM-1) and pro-inflammatory cytokines (IFN-α/β and IL-1) partially drive the clonal-like expansion of NK cells. Inhibitory KIR2DL2/3 in humans and Ly49A/C/G2 in mice can block the membrane proximal activation events upon the recognition of MHC class I.

**Figure 4 cells-10-01955-f004:**
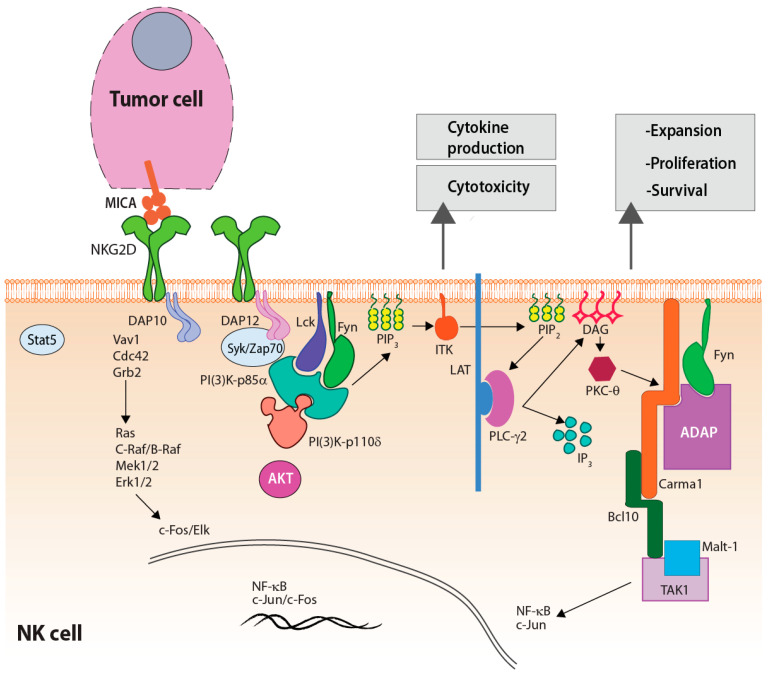
NK cells receive primary signaling through germline-encoded receptors. NKG2D engagement with MHC class I chain related-proteins A (MICA) induces activation of either DAP10 or DAP12 signaling. DAP10 signaling leads to tyrosine phosphorylation of the YINM motif and stimulation of Vav1/Cdc42/Grb2/PI(3)K signaling cascade and nuclear translocation of c-Fos/Elk transcription factors. DAP12 signaling leads to phosphorylation of the ITAM, which allows for Syk/Zap70 signaling cascade to proceed (cytotoxicity and cytokine production).

**Figure 5 cells-10-01955-f005:**
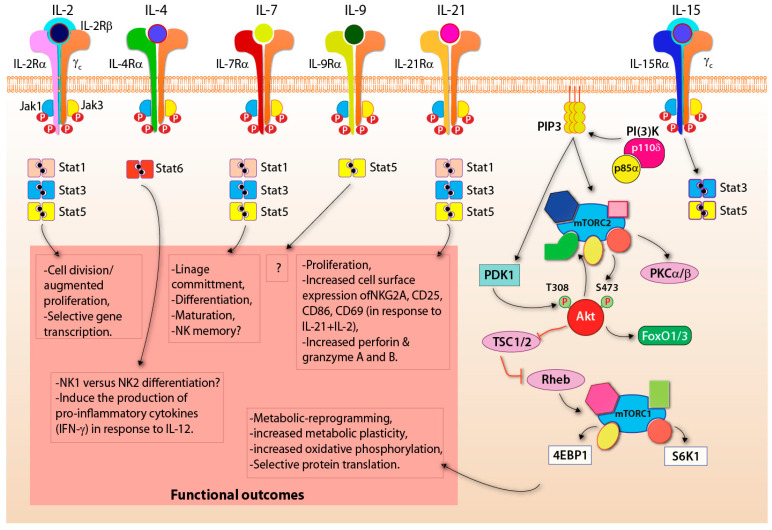
Activation via common γcR constitutes ‘secondary signaling’. Downstream signaling of the common gamma receptor (γcR) family is shown. IL-2, IL-4, IL-7, IL-9, IL-21, and IL-15 activate Janus Kinases 1/3 (JAK), which undergo receptor-mediated phosphorylation. Signal transducer activating transcripts (STAT) are recruited and phosphorylated. Dimerization of phosphorylated STAT translocates to the nucleus to promote various functional outcomes. Also, IL-15 can signal via PI(3)K promoting the conversion of phosphatidylinositol 4,5-bisphosphate (PIP_2_) to phosphatidylinositol 3, 4,5-bisphosphate (PIP_3_), which activates the mammalian target of rapamycin complex 1 (mTORC1) signaling cascade.

**Figure 6 cells-10-01955-f006:**
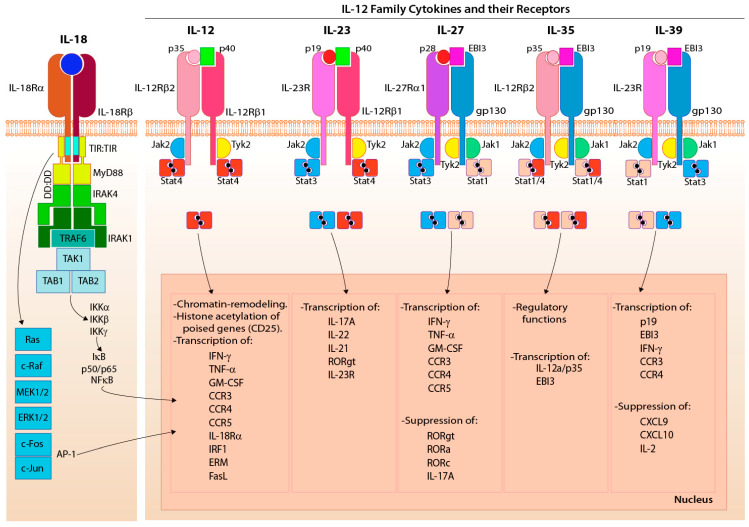
NK cell ‘third signal’ occurs via IL-12 family cytokines and their receptors. Schematic of downstream signaling of the IL-12 family cytokines with their respective transcriptional outcomes. IL-12 consists of p35 and p40 subunits, that activate the IL-12 receptor leading to JAK2/Tyk2 phosphorylation and phosphorylation of Stat4, which can then dimerize and enter the nucleus. IL-23 is composed of p19 and p40 subunits, which activate the IL-23 receptor leading to phosphorylation of JAK2/Tyk2 and signaling via STAT. IL-27 is composed of p28 and Epstein-Barr Virus-Induced 3 (EBI3) subunits and activates the IL-23 receptor leading to phosphorylation of JAK1/2 and Tyk2 and signaling via STAT. IL-35 is composed of p35 and EBI3 subunits and activates the IL-35 receptor leading to phosphorylation of JAK1/2 and Tyk2 and STAT signaling. IL-39 is composed of p19 and EBI3 and activates the IL-35 receptor leading to STAT signaling. Finally, IL-18R does not belong to the IL-12R family but plays a vital role in activating NK cells when co-stimulated with IL-12. Activation via IL-18R leads to the MyD88-dependent signaling pathway.

**Figure 7 cells-10-01955-f007:**
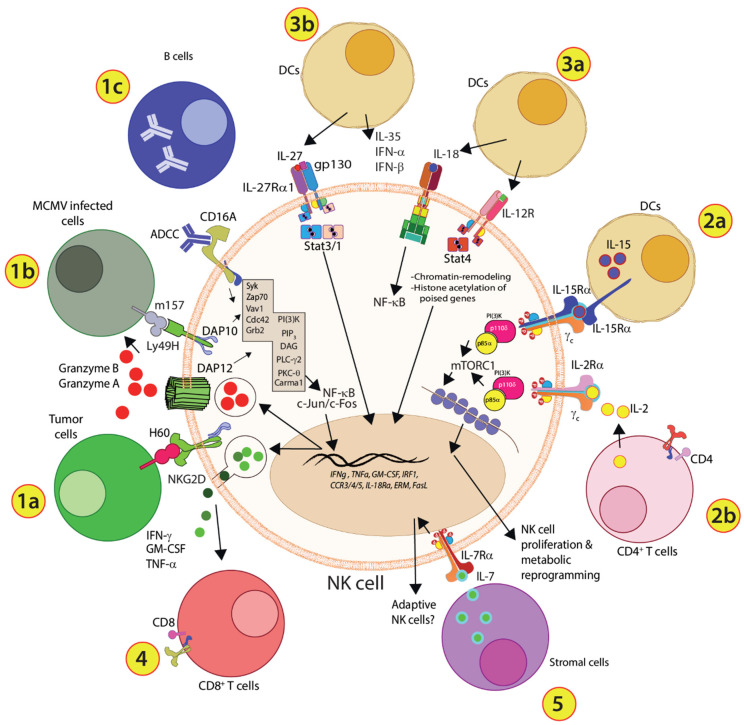
Integrated view of the ‘third signal’ in NK cells. (**1a**) Recognition of tumor cells via stress-induced ligands, such as H60, provides primary activation via NKG2D in murine NK cells. (**1b**) Similarly, recognition of viral proteins including MCMV-derived m157 mediates stimulation of murine NK cells via ly49H. These two activation receptors mediate their signals utilizing DAP10 and DAP12. (**1c**) NK cells also mediate target cell recognition using antibodies bound to its receptor CD16a. The downstream signaling cascades lead to cytokine production and cytotoxicity against target cells. (**2a**) IL-15 produced by DCs is trans-presented to NK cells to proliferate. (**2b**) IL-2 from CD4^+^ T cells plays a similar role in the expansion of NK cells. However, the context where DCs or CD4^+^ T cells play this role is yet fully defined. (**3a**) IL-12 and IL-18 generated by DCs can activate NK cells utilizing a distinct signaling cascade. However, this is thought to occur primarily during viral or bacterial infections. The in vivo context where IL-12 functions, along with NK cell-activating receptors is not known. Irrespective of this, given that Stat4 mediates chromatin remodeling and histone acetylation following IL-12 stimulation indicates that the primary mechanism is to keep the poised genes in the open conformation and, therefore, the parallel stimulation with activating receptors or IL-18 can promote the production of cytokines or cytotoxicity associated proteins. (**3b**) IL-27 and IL-35 produced by DCs also activate NK cells. Signaling by Il-27 is context-dependent and can both activate or inhibit the production of IFN-γ. (**4**) NK cells can activate CD8^+^ T cells via IFN-γ, GM-CSF, or TNF-α. (**5**) IL-7 may play an essential role in promoting NK cells to differentiate into adaptive NK cells.

## Data Availability

No data presented in this review article.

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
