# Peer review of "Implications of a ‘Third Signal’ in NK Cells"

_cells, 2021, doi:10.3390/cells10081955_

Round 1
Reviewer 1 Report
This is an interesting and extensive review that mainly focuses on cytokine receptors as a source of a 3rd signal that regulates NK cell effector function accompanied by 6 colourful figures to illustrate these concepts.
I would recommend the following changes be made to the manuscript prior to publication:
Line 51 – specifically which distant T cell cousins do NK cells resemble?
Line 60/Fig 1. – soluble factors. Growth factors, such as PDGF-D binding to NKp44/NCR2 (Barrow et al. Cell, 2018) and soluble NKG2D ligands (Deng et al., Science, 2015) are neglected, but these are also important soluble regulators of NK cell activation and should be mentioned as 3rd signals, a well as highlighting relevant interactions with stromal cells.
Line 321 – Change ‘IL-15 remains to be a crucial cytokine required for NK cell development, survival, and proliferation.’ to ‘IL-15 is a crucial cytokine..’
Lin 369 – Change subtitle ‘Third signal’ for NK cells: why are they unique? To
Why are ‘Third signals’ unique for NK cells?
Line 579 onwards – change ‘Does the IL-12 family cytokines are the only ones capable of mediating the ‘third signal’?’ to ‘Are IL-12 family cytokines the only ones capable of mediating the ‘third signal’?’
Line 581. The authors ask an interesting question that I feel should be discussed in the review – what is the role of MHC class I in regulating these third signal?
The important MHC-I/KIR or MHCC/Ly49 regulatory interaction should also be illustrated in one of the figures, such as Fig 2?
This is an extensive, well written and engaging review, but the authors should double check overall grammar prior to publication.
Author Response
--We thank the Reviewer for constructive comments. Our answers follow '--' symbol.
This is an interesting and extensive review that mainly focuses on cytokine receptors as a source of a 3rd signal that regulates NK cell effector function accompanied by 6 colorful figures to illustrate these concepts.
--Thank you.
I would recommend the following changes be made to the manuscript prior to publication:
Line 51 – specifically which distant T cell cousins do NK cells resemble?
--We have corrected this to ‘CD8+ T cell’
Line 60/Fig 1. – soluble factors. Growth factors, such as PDGF-D binding to NKp44/NCR2 (Barrow et al. Cell, 2018) and soluble NKG2D ligands (Deng et al., Science, 2015) are neglected, but these are also important soluble regulators of NK cell activation and should be mentioned as 3rd signals, a well as highlighting relevant interactions with stromal cells.
--Thank you and we have included them on page 4.
Line 321 – Change ‘IL-15 remains to be a crucial cytokine required for NK cell development, survival, and proliferation.’ to ‘IL-15 is a crucial cytokine..’
--Thank you and we have changed this sentence.
Lin 369 – Change subtitle ‘Third signal’ for NK cells: why are they unique? To
Why are ‘Third signals’ unique for NK cells?
Line 579 onwards – change ‘Does the IL-12 family cytokines are the only ones capable of mediating the ‘third signal’?’ to ‘Are IL-12 family cytokines the only ones capable of mediating the ‘third signal’?’
--Thank you and the changes are made.
Line 581. The authors ask an interesting question that I feel should be discussed in the review – what is the role of MHC class I in regulating these third signals?
--Thank you and we have added a brief discussion on this on Page 18.
The important MHC-I/KIR or MHCC/Ly49 regulatory interaction should also be illustrated in one of the figures, such as Fig 2?
--We have added this to Figure 2.
This is an extensive, well-written, and engaging review, but the authors should double-check overall grammar prior to publication.
--Thank you and we have added additional corrections.
Reviewer 2 Report
The article by Khalil et al. is a well-written review describing the various types of signals that can lead to the activation of an NK-mediated response. The authors distinguish a primary, secondary or tertiary signal, making a parallel with T lymphocytes.
The strength is the mostly very good language.
Some criticisms:
- Authors should pay more attention to underlining whether some sentences refer to studies carried out in humans or in mice. Indeed, the authors often mix results referring to murine receptors with others referring to human receptors without distinguishing. In some sentences this may not be clear to a non-expert reader (i.e. NKG2D association with DAP12 in mice and DAP10 in humans; association of NKG2B with HLA-E demonstrated in mice but not in humans; Ly49 in mice and KIR in humans).
- In the paragraph “Germline-encoded receptors form the conduits of ‘primary signal’ in NK cells” please add more details on NCRs. NCRs are the main activating receptors of human NK cells. The stimulatory receptor NKp46/NCR1 is the first identified NCR (Sivori et al J. Exp. Med. 1997; Pessino et al J. Exp. Med. 1998). It efficiently triggers the release of cytotoxic granules, cytokines, and chemokines upon binding ligands of viral (Mandelboim et al Nature 2001), bacterial (Vankayalapati et al, J. Immunol. 2002), and cellular origin (Narni-Mancinelli et al., Sci. Immunol. 2017) in addition to unidentified ligands on tumor cells. The mouse ortholog of NKp46/NCR1, termed MAR-1, has been identified (Biassoni et al., Eur. J. Immunol. 1999). Please add more details in the text.
- Line 83: In addition to NCR1 the authors should also add NCR3 as it is able to induce a potent activation of NK cells. Some of its ligands have also been defined. Authors should add and comment on these important concepts.
- NCR1 and Ly49 are very different receptors: NCR1 is a human non-MHC-specific receptor while Ly49 are murine MHC-specific receptors. It would be better to separate them into 2 separate paragraphs. This would allow to the authors to better describe the NCRs, their ligands and their important role. Moreover, by moving NCR1 from this paragraph to another, the authors could better focus their attention on the parallelism between Ly49H and NKG2C, both MHC-specific receptors and involved in the response to CMV. Moreover, please add more details on human adaptive NK cell subset.
- Primary signals can be induced also by tumor cells; authors could add a paragraph on this issue.
- Line 107: Please specify inhibitory KIRs.
- Line 187: please change to “ADCC by NK cells is mainly mediated by IgG-subclasses IgG1 and IgG3 through CD16A.”
- Human NK cells express different types of TLR, not only TLR2. Please add something about TLR3, TLR9 expression and function in human NK cells (Sivori et al Proc Natl Acad Sci U S A. 2004)
Author Response
--We thank the Reviewer for constructive comments. Our answers follow '--' symbol.
The article by Khalil et al. is a well-written review describing the various types of signals that can lead to the activation of an NK-mediated response. The authors distinguish a primary, secondary, or tertiary signal, making a parallel with T lymphocytes.
The strength is the mostly very good language.
Some criticisms:
Authors should pay more attention to underlining whether some sentences refer to studies carried out in humans or in mice. Indeed, the authors often mix results referring to murine receptors with others referring to human receptors without distinguishing. In some sentences, this may not be clear to a non-expert reader (i.e. NKG2D association with DAP12 in mice and DAP10 in humans; association of NKG2B with HLA-E demonstrated in mice but not in humans; Ly49 in mice and KIR in humans).
--Thank you for the suggestion. In the current version of the manuscript, we have indicated mouse and human in multiple places, including the ones suggested by the reviewer.
In the paragraph “Germline-encoded receptors form the conduits of ‘primary signal’ in NK cells” please add more details on NCRs. NCRs are the main activating receptors of human NK cells. The stimulatory receptor NKp46/NCR1 is the first identified NCR (Sivori et al J. Exp. Med. 1997; Pessino et al J. Exp. Med. 1998). It efficiently triggers the release of cytotoxic granules, cytokines, and chemokines upon binding ligands of viral (Mandelboim et al Nature 2001), bacterial (Vankayalapati et al, J. Immunol. 2002), and cellular origin (Narni-Mancinelli et al., Sci. Immunol. 2017) in addition to unidentified ligands on tumor cells. The mouse ortholog of NKp46/NCR1, termed MAR-1, has been identified (Biassoni et al., Eur. J. Immunol. 1999). Please add more details in the text.
--Thank you for all these suggestions and references. We have added this information in the current version of the manuscript.
Line 83: In addition to NCR1 the authors should also add NCR3 as it is able to induce a potent activation of NK cells. Some of its ligands have also been defined. Authors should add and comment on these important concepts.
--Thank you and we have added information regarding NCR3 in the current version of the manuscript.
NCR1 and Ly49 are very different receptors: NCR1 is a human non-MHC-specific receptor while Ly49 is murine MHC-specific receptors. It would be better to separate them into 2 separate paragraphs. This would allow to the authors to better describe the NCRs, their ligands and their important role. Moreover, by moving NCR1 from this paragraph to another, the authors could better focus their attention on the parallelism between Ly49H and NKG2C, both MHC-specific receptors and involved in the response to CMV. Moreover, please add more details on the human adaptive NK cell subset.
--Thank you for this constructive suggestion. We appreciate and have done this in the current version of the manuscript.
Primary signals can be induced also by tumor cells; authors could add a paragraph on this issue.
--This is covered in Section 2.1. Thank you.
Line 107: Please specify inhibitory KIRs.
--Thank you and it is included.
Line 187: Please change to “ADCC by NK cells is mainly mediated by IgG-subclasses IgG1 and IgG3 through CD16A.”
--Thank you, and we have incorporated this in the text.
Human NK cells express different types of TLR, not only TLR2. Please add something about TLR3, TLR9 expression and function in human NK cells (Sivori et al Proc Natl Acad Sci U S A. 2004)
--We have included detailed information on which TLR is expressed on human and murine NK cells and the functions they perform. Please see page 11 of the current version of the manuscript.